## RESEARCH ARTICLE

# Thermal window of exercise performance of the ecosystem engineer *Lanice conchilega*

Nienke Zwaferink[1,2], Paula de la Barra[1,*] and Katharina Alter[1,‡,§]

## ABSTRACT

Ocean warming is reshaping marine ecosystems and shifting species distributions. Resilient habitat-forming species help stabilize conditions for other organisms, supporting community structure under change. The tube-worm *Lanice conchilega* is such a habitat-former, enhancing species richness in sandy environments. Its thermal performance range remains unknown, partly because standard methods are poorly suited for this species. We present a new experimental approach to assess thermal performance based on tube-building activity, an important trait for physical protection, feeding, and habitat engineering. Spring-collected individuals were exposed in the laboratory to an ecologically relevant temperature range. Tube-building activity matched spring field conditions with a thermal minimum, optimum, and maximum at 3.6, 12.4, and 21.4°C, respectively. Performance depended strongly on recent thermal history. Because thermal tolerance can shift through acclimation, seasonal performance curves are needed to determine whether cold winters or hot summers may constrain this ecosystem engineer with potential consequences for intertidal community structure.

KEY WORDS: Polychaete worm, Stress tolerance, Functional performance, Thermal performance curve, Habitat-forming species, Tube-building activity

## INTRODUCTION

As climate change and other anthropogenic pressures make environments harsher for many species (Pecl et al., 2017), facilitation is becoming an increasingly important process for communities worldwide (He et al., 2013). Resilient habitat-forming species, capable of alleviating stressful conditions for numerous other species, play a vital role in sustaining their communities during the current biodiversity crisis (Bulleri et al., 2018). To maintain ecosystem resilience, habitat-forming species are essential. In coastal habitats, these are often mollusks, crustaceans, and polychaetes but for some of

[1]Department of Coastal Systems, Royal Netherlands Institute for Sea Research, PO Box 59, 1790 AB Den Burg, The Netherlands. [2]Department of Marine Biology, Groningen Institute for Evolutionary Life Sciences, University of Groningen, Nijenborgh 7, 9747 AG Groningen, The Netherlands.
*Present address: UK Center for Ecology and Hydrology, Bangor, United Kingdom.
‡Present address: Department of Marine Biology, Institute of Biological Sciences, University of Rostock, Rostock, Germany.

§Author for correspondence (katharina.alter@uni-rostock.de)

N.Z., 0009-0000-3181-1621; P.d.l.B., 0000-0001-8416-5926; K.A., 0000-0002-1801-748X

these species their ability to cope with future climate-driven changes remains uncertain.

Temperature strongly influences the physiology and ultimately the distribution of ectothermic species (Sunday et al., 2012). In temperate environments, especially intertidal species experience large temperature fluctuations due to tides, weather, oceanic and seasonal dynamics. For instance, in the North Sea's tidal system, the Wadden Sea, average sea surface temperatures vary between 0.4 and 6.9°C in winter and increase by about 8°C between late March and late June, with extremes ranging from 2 to 23°C (Royal Netherlands Institute for Sea Research, www.dataverse.nioz.nl, years 2001–2021).

The population dynamics of the habitat-forming polychaete *Lanice conchilega* (Pallas, 1766) are shaped by climatic fluctuations, particularly conditions experienced during winter (Beukema, 1992; Strasser and Pieloth, 2001). Its abundance is increasing after mild winters, which indirectly enhances local biodiversity (de la Barra et al., 2025). By constructing dense tube aggregations from sand and shell fragments, *L. conchilega* modifies sediment dynamics, nutrient fluxes, and habitat structure, which results in locally increased species richness and faunal abundance (Forster and Graf, 1995; Rabaut et al., 2007; Van Hoey et al., 2008; de la Barra et al., 2025). Determining the thermal sensitivity of *L. conchilega* is important for understanding how temperature affects its functioning and the resulting implications for macrozoobenthic intertidal communities.

However, no data exist on the thermal performance range of *L. conchilega*, partly because standard methods are not applicable. Thermal performance curves describe how temperature affects biological function, from cellular processes to whole organism and population responses (Rezende and Bozinovic, 2019). Although performance curves vary in shape, they generally increase from a thermal minimum, below which physiological function ceases, to peak at the thermal optimum, where maximum performance occurs. With further increasing temperatures, performance typically declines sharply until the thermal maximum is reached, beyond which animals can no longer function (Arnoldi et al., 2025). Conventional techniques used to generate performance curves, such as oxygen consumption rates (Wittmann et al., 2008), movement or activity rates (Schröer et al., 2009), and vital rates such as growth and reproduction (Lewis et al., 2002), are difficult to apply to *L. conchilega*. This species is an obligate tube dweller that does not survive outside its sedimentary structure, yet the tube's properties complicate both physiological and behavioral measurements. For example, the tube is permeable to oxygen, and the worm exhibits irregular oxygen-pumping behavior (Forster and Graf, 1995), which prevents reliable respirometry. Behavioral observations are also constrained, as the worm remains hidden within its tube, and the lack of established rearing techniques further limits assessments of development and growth.

In this study, we present a novel experimental approach for assessing thermal performance in *L. conchilega* based on its tube-building activity, a trait essential for building a physical protection, to form a structure important for feeding, and habitat engineering

(Fig. 1). Spring-acclimatized individuals were collected from the intertidal and exposed for 20 h to a range of ecologically relevant temperatures (5–23°C). Measuring tube-building capacity provided an ecologically meaningful proxy for evaluating the thermal performance curve in this otherwise methodologically challenging species. Because our experiment was conducted with spring-acclimatized individuals, it does not allow direct inference about the species' winter sensitivity (de la Barra et al., 2025). Instead, our study establishes a method for quantifying how *L. conchilega* responds to short-term temperature variation. It provides a baseline understanding of its temperature-dependent tube-building ability under spring conditions that can be extended to other seasons.

## RESULTS
### Field temperature
In the field, average temperatures were similar at the sediment surface compared to temperatures at 20 cm sediment depth with values of 5.0 and 5.4°C during the coldest month of the sampling period, i.e. January, 10.8 and 10.6°C in April, which was the month prior experiments, and 20.3 and 19.6°C during the hottest month of the sampling period, i.e. June, for sediment surface and 20 cm depth, respectively (Fig. 2). Yet, the sediment buffered temperature fluctuation by 9–10°C. At the sediment surface temperatures varied by 13.5°C (−1.8 to 11.7°C), 12.2°C (5.3 to 17.5°C), and 17.2°C (11.9 to 29.1°C) in January, April, and June, respectively, while at 20 cm sediment depth temperature fluctuations were reduced to 4.3°C (2.3 to 6.6°C), 3.0°C (9.3 to 12.3°C), and 7.4°C (15.1 to 22.5°C) in January, April, and June, respectively (Fig. 2).

### Biometric characteristics and thermal performance
Body size ranged from 42.6 to 153.9 mg dry weight (DW) with an average of 92.0±3.1 mg DW (mean±s.e.m.) and a median of 94.1 mg DW (Fig. 3A,B). The maximum newly built tube segment was 17.0 cm long and was constructed by an individual exposed to 11°C. Out of 114 individuals used, 15 did not build a new tube segment, with five at 5°C, two at 19°C, two at 21°C, and six at 23°C. Out of eight individuals exposed to 23°C, six expelled themselves from their tubes and only one built a new tube segment. See Fig. 3 for examples of old and new build tube segments.

The Gaussian model fitted significantly better to the data compared to the other tested models (ΔAICc=3) and estimated a thermal optimum of 12.4°C (95% CI: 11.9–13.1°C) with a corresponding maximum new tube length of 8.9 cm (95% CI: 7.3–11.0 cm) (Table S1, Fig. 4). The thermal minimum and maximum were estimated at 3.6°C (95% CI: 1.8–6.3°C) and 21.4°C (95% CI: 19.6–22.9°C), respectively (Table S1, Fig. 4).

## DISCUSSION
This study provides the first experimentally derived thermal performance curve for *L. conchilega* using an exercise performance proxy, i.e. tube-building activity, to determine organismal functioning across an ecologically relevant temperature range. We used spring-acclimatized individuals and their thermal performance closely matched field conditions prior to the month of collection. Tube-building activity ceased 1.7°C below the measured minimum sediment surface temperature recorded in the field, peaked at 1.6°C above the average temperature, and declined markedly at temperatures exceeding the maximum field temperature, with the first individuals failing to build tube segments at temperatures 1.5°C above the maximum. Together these results suggest a strong dependence of tube-building performance on recent thermal history and indicate that the thermal coping range measured here reflects

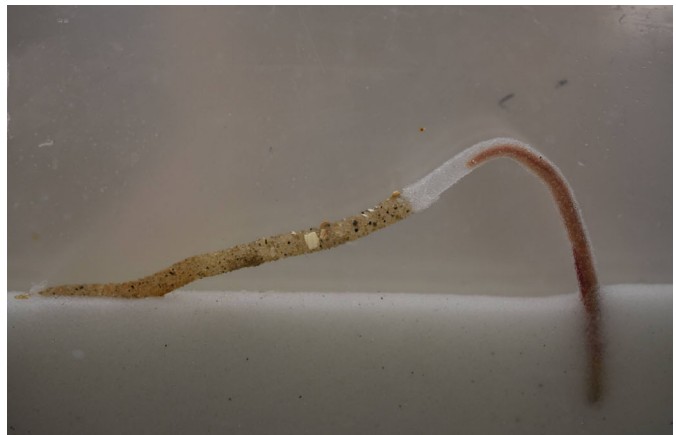

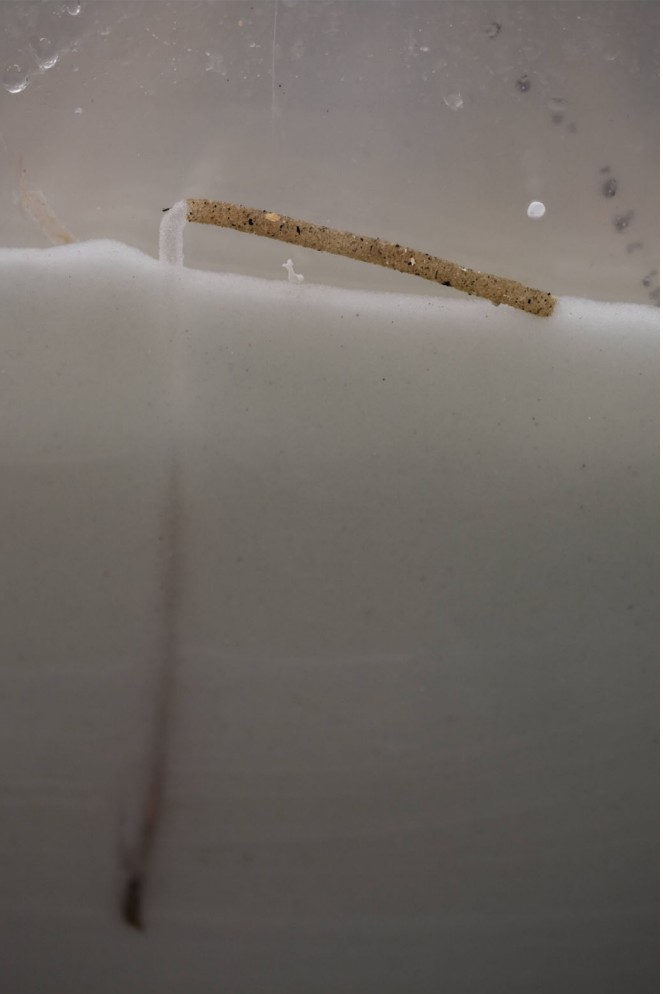

**Fig. 1. Example images from the experiment.** Extending from its 7 cm long natural (darker) tube located on the surface of transparent glass beads, individual *L. conchilega* (visible in red) re-built tubes from glass beads several centimeters long in narrow (0.6 cm wide) aquaria.

spring acclimation rather than a fixed threshold for this species that lives in temperate habitats.

Sediment surface temperatures can approach 0°C and occasionally drop below this temperature (minimum measured in this study: −1.8°C) in the Wadden Sea in winter. *Lanice conchilega* is considered a winter-sensitive species and previous studies demonstrated a link between mild winters and subsequent

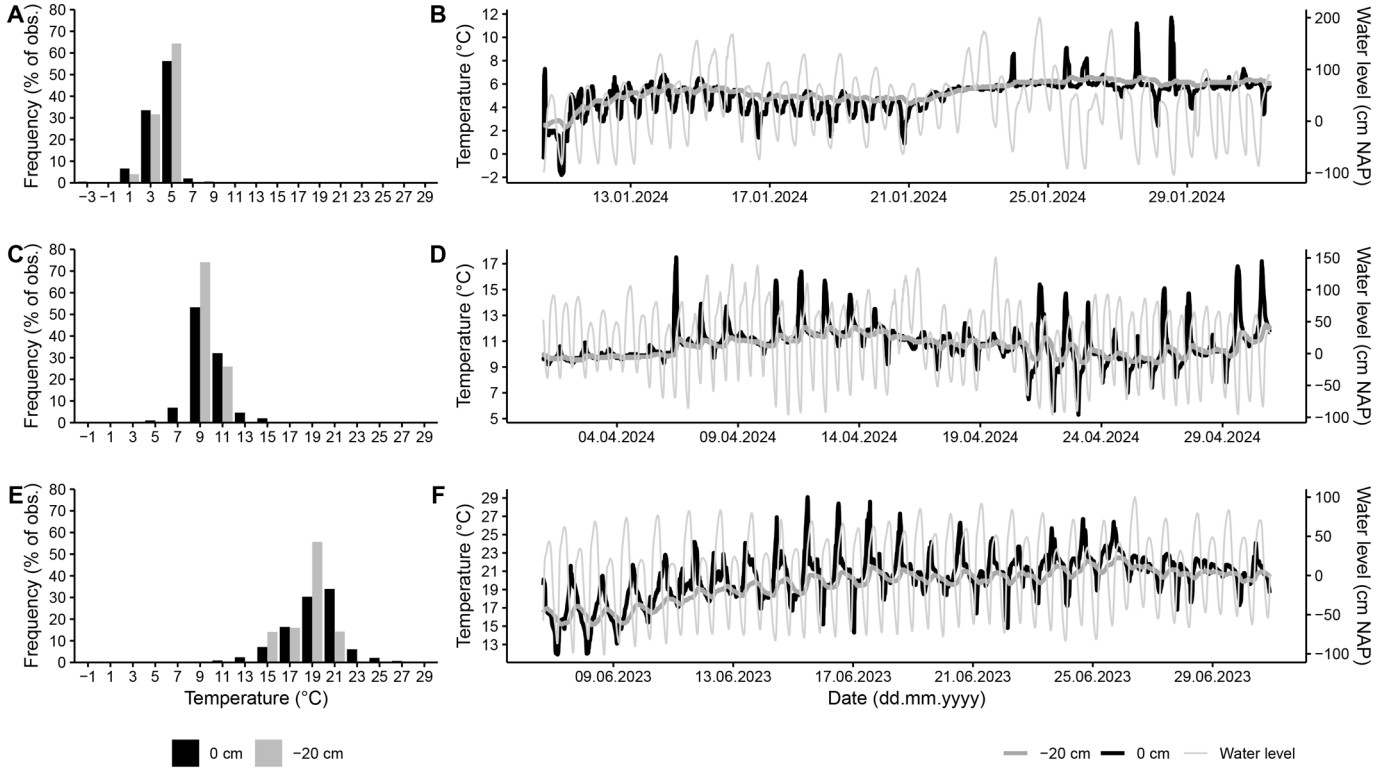

**Fig. 2. Intertidal mudflat temperatures.** Temperature (°C) was logged continuously at a rate of one reading per 5 min at the sediment surface [black bars (A,C,E) and black lines (B,D,F)] and in 20 cm sediment depth [grey bars (A,C,E) and thick grey lines (B,D,F)]. Data are shown as frequency distribution (A,C,E) and in detail across the tidal cycle indicated as water level [(cm above Normaal Amsterdams Peil (NAP)]; (B,D,F) across the coldest month of the year (January, A, B), the sampling month of *L. conchilega* (April, C, D), and the warmest month of the year (June, E, F). obs., observations.

population increases during the following summer (de la Barra et al., 2025). It has been suggested that *L. conchilega* may reduce the risk of overwinter mortality by retracting deeper into their tubes, and with that into somewhat warmer sediment layers, similarly to other benthic infauna during cold stress (Masanja et al., 2023; de la Barra et al., 2025). Indeed, vertical positioning in the sediment would reduce cold stress of *L. conchilega* as temperature at 20 cm depth rarely dropped below the thermal minimum of the species during our study period. Yet, the lower thermal limit determined in this study cannot be directly interpreted as evidence that winter temperatures constrain *L. conchilega* in nature. Many ectotherms seasonally adjust

their thermal limits through phenotypic plasticity to enhance performance at the prevailing temperatures (Hopkin et al., 2006; Schröer et al., 2011; Hahn and Brennan, 2024). If *L. conchilega* similarly acclimates to colder conditions, its capacity to maintain tube structure and ultimately survive and stabilize population dynamics, may be considerably greater than suggested from measurements determined in spring. To assess winter vulnerability of *L. conchilega*, thermal performance should be measured in winter-acclimatized individuals.

A similar argument applies to the upper thermal limit determined in the present study. Tube-building activity declined markedly

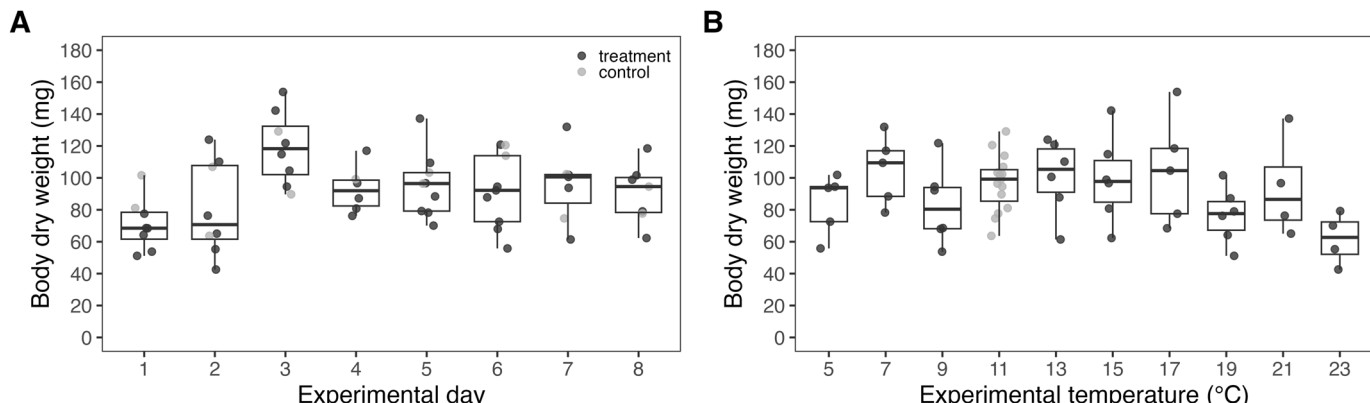

**Fig. 3. Biometric measurements of *L. conchilega*.** Body dry weight (mg) of control (11°C, grey circles) and treated (black circles) individuals used across experimental days (A) and experimental temperatures (°C; B). Boxplots display the interquartile range (IQR; 25th–75th percentiles), with whiskers extending to the most extreme values that fall within 1.5×IQR of the lower and upper quartiles. Closed circles beyond this range are plotted as outliers. The median is indicated by a horizontal line within each box. Closed circles represent the number of aquaria that were used across experimental days and temperatures. Each aquarium contained two *L. conchilega* individuals whose performance was averaged.

Biology Open

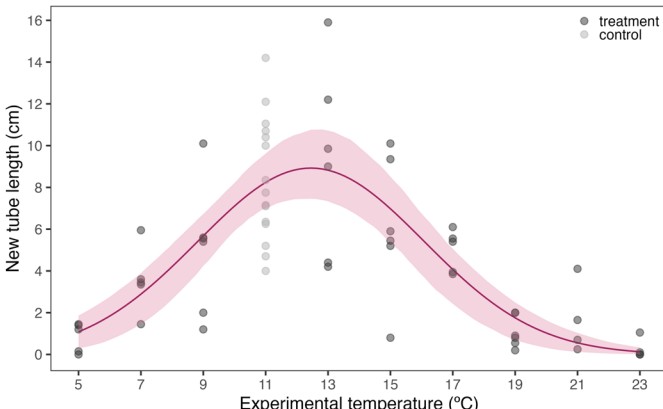

**Fig. 4. Tube building activity (cm) of *L. conchilega* across temperatures (°C).** Tube length was measured after 20 h of temperature exposure. The shaded area represents the 95% bootstrapped confidence interval around the mean (solid line) predicted by a Gaussian model. Closed circles (*n*=4–15) represent the number of aquaria that were used for each temperature. Each aquarium contained two *L. conchilega* individuals whose performance was averaged.

above 17°C and most individuals showed acute heat stress at 23°C by abandoning their tube. However, summer sediment temperatures in the intertidal Wadden Sea can reach 29°C, suggesting that the upper thermal limit must increase during the warmer months of the year. Given that the species' geographic distribution extends to the Mediterranean Sea and the Gulf of Mexico (OBIS, 2025), where typical summer temperatures exceed those of the North Sea, it is likely that the species can tolerate higher temperatures than those measured in spring in this study. Repeating our experiment with individuals collected in summer would allow us to test if *L. conchilega* also shifts its thermal window, similar to other intertidal polychaetes from European Seas (Wittmann et al., 2008; Schröer et al., 2011).

Measuring tube-building activity proved to be a possible proxy for exercise performance in a species for which traditional physiological approaches were less suitable. The technique captured behavioral responses to temperature while preserving a natural sedimentary context. Another exercise performance, i.e. burrowing rate, has been used before to determine thermal sensitivities in a marine polychaete, *Arenicola marina* (Linnaeus, 1758), sampled in different seasons and latitudes (Schröer et al., 2009, 2011). While lower critical temperatures were not identified even during exposure to −1°C, the thermal optimum of *A. marina* acclimated to 10°C was 13.3°C for individuals from the French Atlantic intertidal and 14.5°C for individuals from the North Sea and the upper critical temperature was 27.0°C for North Sea individuals (Schröer et al., 2009, 2011). The thermal optimum of *L. conchilega* determined in the present study is comparable to that of *A. marina*, yet the loss of tube-building capacity at the most extreme tested temperatures and therefore narrower thermal range indicates a higher thermal sensitivity for exercise performance in *L. conchilega*.

Our approach has several limitations that must be considered when interpreting our results. Individuals were collected from the field during low tide, immediately transported to the laboratory, and used in experiments commencing on the same day. We cannot rule out that the potential handling stress impacted the tube building performance, e.g. by depleting short-term energy reserves. Tube construction requires mucus secretion for building of an inner organic layer and burrowing into the sediment (Ziegelmeier, 1952). Consequently, it also depends on muscular activity and energy capacity to sustain it. Additionally, tube-building is an integrated measure of organismal function and other vital processes, e.g.

feeding or metabolism, may have different thermal sensitivities as shown for other ectotherms (Monaco et al., 2017; Eymann et al., 2020; Alter et al., 2025). The thermal tolerance curve presented here thus captures one ecologically important function but not the full thermal landscape of the species.

Despite these limitations, our study provides a baseline for how temperature affects tube-building performance in *L. conchilega*. We argue that measuring tube-building ability is a valid proxy for organismal functioning because tube construction is fundamental for *L. conchilegas*' protection, feeding, and burrow stability and it also mediates its ecosystem-engineering effects by maintaining habitat complexity and sediment stabilization. However, because thermal tolerance can shift seasonally through acclimation, performance curves from different seasons must be compared in order to determine whether the species could be constrained by cold conditions in winter or by extreme warming in summer. Given its role as an ecosystem engineer, shifts in *L. conchilega* abundance or spatial distribution due to shifts in temperature have the potential to reshape intertidal community structure.

## MATERIALS AND METHODS
### Field temperature
To characterize the temperature fluctuations to which the animals are exposed to in their intertidal habitat, temperature loggers (EnvLoggers T7.3, ElectricBlue, Portugal) were placed within the high intertidal zone of the Wadden Sea coast off the island Texel, the Netherlands (53°00614″N, 4°784917″E) at the sediment surface and at 20 cm sediment depth. Temperature was logged continuously at a rate of one reading per 5 min from May until November 2023 and from January until June 2024.

### Species collection
*Lanice conchilega* individuals were collected from the intertidal area northeast of the Wadden Sea island Texel, the Netherlands (53°10'00.0″N, 4°52'30.7″E). Collection took place for eight consecutive days during low tide in late April and early May 2024. Animals were collected within their tubes and transported to the laboratory in seawater at the local environmental conditions (11°C). The tube lengths varied in size and one side had a fringe of variable size that collapsed during collection. To standardize tube size and structure at the start of the experiment, we cut all tubes to 7 cm whereby care was taken to not harm the individual inside the tube. The tube length was large enough for the worms to hide in and this procedure ensured consistent starting conditions. Individuals were used in experiments commencing on the same day.

### Experimental design
The thermal performance of *L. conchilega* was determined by exposing different individuals for 20 h to one of ten test temperatures ranging from 5 to 23°C at 2°C intervals. After the temperature exposure, the length of newly built tube was measured (Fig. 1). For this, two individuals were placed horizontally and with their anterior sides facing one another on a layer of artificial sand (glass beads 200–300 μm diameter) in an aquarium (40×35×0.6 cm, L×H×W, *n*=6–9 per temperature). Each aquarium contained a 32 cm-deep artificial sand layer, which extended to 3.5 cm below the seawater surface, ensuring that animals remained fully submerged. The aquarium was connected to a flow-through seawater system with the inflow placed in the center of the two *L. conchilega* individuals and the outflow on both narrow sites of the aquarium. Aerated, filtered (1.0, 0.5, 0.2 μm, active charcoal), and UV-treated (55-watt UV-C lamp) local seawater was used with a flow rate of 8 liters h$^{-1}$. Aquaria (*n*=1–3 per temperature and experimental day) were placed in water baths (*n*=4) set to four different temperatures inside a temperature and light-controlled room (Table S2). The control temperature (local condition, 11°C) was used in one water bath each day to account for potential differences across experimental days (*n*=8, Table S2). All experimental temperatures were tested in two or three experimental days so that any differences in condition of individuals between days was randomized across temperatures (Table S2). Water temperature was controlled by heaters (100-600 watt, TR2 controllers,

Biology Open

Schego, Germany) and coolers (TK20000H, TECO, Italy, and ALPHA RA 24, LAUDA, Germany) and was measured in the water baths every 10 min (HOBO Pendant MX Water Temperature Data Logger MX2201) and in the sediment before and after each experimental run (GM1312 Digital Thermocouple Thermometer, Walfront, USA).

For the duration of each experimental run (20±0 h), individuals were left undisturbed with continuous red lighting (511046B0, Eurolite, Germany). After 20 h, tubes (original part with new built tube) were removed from the aquaria. Individuals were expelled from tubes, assessed for mortality by visible inspection of movement, and stored at −80°C until freeze-drying (−30°C, 0.2 mbar, Telstar LyoQuest-55, Spain) took place to determine DW. The original tube (7 cm) was distinguishable from the newly built segment due to a color contrast between the darker natural material and the lighter artificial sand (Fig. 1). Digital images of the tubes were taken (12MP iPhone SE, 2022) for length determination using image analysis (ImageJ, FIJI). Tubes were dried at room temperature for 11–18 days and the weights of the original and newly built tube were measured (0.1 mg precision, Mettler Toledo XS204, Australia).

Out of 126 individuals used, 12 were excluded from the analyses due to experimental or biological constraints. They either did not show any signs of movement and were considered dead, their soft tissue was injured, they were identified as a different species, their body weight could not be determined, or their tube was misplaced, which prevented artificial sediment access to the individual.

## Analyses

Temperatures measured in the field were used to construct frequency histograms of thermal exposure during the coldest and warmest month of the year, as well as during the month preceding the experiments.

For the 114 individuals included in the analysis, tissue DW and newly built tube length were averaged across the two individuals per aquarium to account for pseudo-replication (Hurlbert, 1984). Newly built tube length was not standardized for individual size because body mass was determined only after the experiments. Because experimental temperature may influence body mass, mass could not be considered independent of treatment. Newly built tube length was analyzed as the cumulative length produced over the experimental period, as experimental duration was identical across treatments and experimental runs (20±0 h; mean±s.d.) and tube building activity was not assumed to be constant over time.

Nonlinear least squares regressions were fit on data of newly built tube length using the *rTPC* package in R v. 4.4.1 (Padfield et al., 2021; R Core Team, 2024). We tested a set of *rTPC* models selected based on the following criteria: the model needed to (1) estimate the thermal optimum, (2) predict positive response variable values, and (3) easily fit biological rate data recommended by Padfield et al. (2021), which yielded four models (Gaussian_1987, Oneill_1972, Pawar_2018, Weibull_1995). The best-fitting model was determined based on the smallest Akaike's Information Criterion corrected for small sample size (AICc) whereby a difference of 2 was considered significant (Burnham and Anderson, 2004). The estimates of the thermal performance parameters, i.e. thermal optimum and the corresponding maximum performance rate, thermal minimum, and thermal maximum, were extracted and their 95% confidence intervals (CIs) were derived by bootstrapping (case resampling) using the *car* package (Fox and Weisberg, 2019).

## Acknowledgements
We thank staff and students from the Royal Netherlands Institute for Sea Research for their help, especially Robert Twijnstra and Scott Maxson for their support in the laboratory as well as Sophie Brasseur, Max Burgoon, Bastiaan van Gemert, Antoine Grenier-Journe, Mario Francesco Tantillo, Koen Stork, and Jara Westbeek, for their assistance during species collection in the field.

## Competing interests
The authors declare no competing or financial interests.

## Author contributions
Conceptualization: K.A., P.d.l.B.; Data curation: N.Z.; Formal analysis: N.Z.; Funding acquisition: K.A.; Investigation: K.A., N.Z., P.d.l.B.; Methodology: K.A., N.Z., P.d.l.B.; Resources: K.A., P.d.l.B.; Supervision: K.A., P.d.l.B.; Validation: K.A.; Visualization: N.Z.; Writing – original draft: K.A., N.Z.; Writing – review & editing: K.A., N.Z., P.d.l.B.

## Funding
This research was supported by HORIZON EUROPE European Research Council research and innovation program HORIZON-CL6-2021-BIODIV-01-04 under grant agreement no. 101060072 "ACTNOW" - Advancing understanding of cumulative impacts on European marine biodiversity, ecosystem functions, and services for human wellbeing". Open Access funding provided by University of Rostock. Deposited in PMC for immediate release.

## Data availability
The data generated and analysed in the present study are available from Zenodo Digital Repository at https://doi.org/10.5281/zenodo.17715259. All other relevant data and details of resources can be found within the article and its supplementary information.

## Peer review history
The peer review history is available online at https://journals.biologists.com/bio/lookup/doi/10.1242/bio.062398.reviewer-comments.pdf

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
