## [Peer Review File · Biology Open]

Thermal window of exercise performance of the ecosystem engineer *Lanice conchilega*

Nienke Zwaferink, Paula de la Barra and Katharina Alter
DOI: 10.1242/bio.062398

Editor: Lewis Halsey

Review timeline

Original submission:	27 November 2025
Editorial decision:	3 December 2025
First revision received:	18 December 2025
Accepted:	19 December 2025

Original submission

First decision letter

MS ID#: bio.062398

MS Title: Thermal window of exercise performance of the ecosystem engineer *Lanice conchilega*

Authors: Katharina Alter; Nienke Zwaferink; Paula de la Barra

Dear Dr Alter,

I have now reached a decision on the above manuscript.

The reviewer reports are shown at the bottom of this email.

As you will see, the reviewers raised a number of substantial criticisms that prevent me from accepting the paper at this stage.

They suggest, however, that a revised version might prove acceptable, if you can address their concerns. If you think that you can deal satisfactorily with the criticisms on revision, I would be pleased to see a revised manuscript. We would then return it to the reviewers.

Reviewer 1

Comments for the author

General remarks:

In this study, the authors generate a thermal performance curve for a species that does not lend itself readily to such investigations owing to its obligatory relationship with sandy sediments. I found the study to be an interesting contribution, especially since there was a nice correspondence between the TPC and the temperature regime to which it is exposed in the field. While further studies on summer and winter acclimated animals would provide for a more complete characterisation of its TPC, the present study already shows that the methodology works for this species. My main suggestion is to do more with the data: Currently, data from two specimens is averaged and the analysis is focussed on tube length, but in addition to tube length, also tube weight and the mass of the animal itself are measured. Since this study is the first on this species it

would perhaps be nice to analyse what performance measure is most appropriate (tube length, tube weight, % new tube mass, etc.) and explore whether the performance metric can be standardized by the mass of the animal.

Detailed remarks:

Line 75: for a study explicitly studying the shape of the thermal performance curve across different levels of biological organisation see:

<https://doi.org/10.1098/rstb.2018.0549>

Line 78: maybe say: "performance typically declines sharply". Note also that the paper cited (Portner's JEB review) (i) shows a more symmetrical unimodal curve, rather than a skewed performance curve which is more typical (e.g. see: <https://doi.org/10.1073/pnas.2513099122>), and (ii) different studies define the thermal maximum differently. Typically, T_{max} is the temperature at which performance becomes 0, and this is frequently equated to the upper thermal maximum, above which the animal dies from heat stress. See also:

<https://doi.org/10.1073/pnas.0709472105>

Line 93: I would say this study evaluates the thermal performance curve, rather than thermal tolerance (which is only the latter part of the thermal performance curve).

Line 144: Why was the average taken from the two individuals, rather than using both as separate data points (with possibly aquarium as a random factor to account for any effects)?

Line 169: I would also include a figure of the temperatures to which these animals are likely exposed. It is highly interesting to see that the thermal range of the TPC is nicely bracketed by the lower and higher temperature ($\sim 7^{\circ}\text{C}$ and $\sim 21^{\circ}\text{C}$).

Line 181: what was it about their physical state that prompted you to exclude said individuals? Also, I would place this information in the methods and focus in the results on those animals that were used on the analysis.

Line 184: given that the mass of both the individual animals and that of the tube was measured, I was expecting an analysis where individual mass was included as a covariate. Also, the use of a control group is a great idea to test for variation in the condition of specimens collected on different days, so I was expecting an explicit test of this (and if there is no bias in the condition, then the control data could be added to the overall analysis).

Line 195: Not sure what the function is of figure 2. Like I mentioned above, I would use an explicit analysis to test for differences across days in performance and then test for variation in tube building performance related to body mass. Then the response variable (mass of newly build tube) could be corrected for differences in mass and then this mass-corrected performance could be fitted with a thermal performance gaussian regression.

Line 216: Not sure that I follow this rationale. The TPC is bracketed by the thermal minimal and thermal maxima currently experienced. So there is some local adaptation, but concluding that these are spring acclimated and thus that the species will show a different TPC when summer acclimated is somewhat speculative with the data at hand. It could be that the species becomes less active during summer, but the temperatures in its habitat are not so extreme that they surpass the thermal range fitted here, right? I guess would could be better highlighted as spring acclimation is the peak in the TPC which coincides exactly with the temperature experienced.

Line 301: The doi link is not working.

Reviewer 2

Comments for the author

This paper aims at testing the effect of temperature on a habitat-forming species, the tube worm *Lanice conchilega*. Because standard methods are unsuitable to measure this species' performance, the authors devised a simple method that takes into account the ability of this species to form their tube. The authors found that tube-building activity peaks at around 12°C , with minimum and maximum at around 3°C and 21°C , respectively. Tube-building by this species is an activity of great ecological importance, since it modifies sediments and habitat structure, and can positively affect species richness. Therefore, by devising a simple methodology to measure tube-building performance and by assessing the temperature performance curve of this key species, this paper provides a potentially useful contribution to the literature on the effect of global warming on intertidal communities. I only have a few minor comments.

Specific comments

1) The figures are clear and the experiments were appropriately designed, including a control. The statistics are appropriate.

2) The authors conclusions are supported by the data. The paper's limitations section is also well written and the use of only one season is listed as one of the limitations. (Line 93). The authors wrote: "Because our experiment was conducted with spring-acclimatized individuals, it does not allow direct inference about the species' winter sensitivity (de la Barra et al., 2025)." This is true for the winter but also of any other season.

3) The state of the art is cited appropriately.

4) A new, simple method to assess performance is carefully described.

5) line 116: "To determine the thermal performance of *L. conchilega*, individuals were exposed to every other temperature between 5 and 23 °C for 20 h." Not clear what temperatures the authors refer to. You mean every other degree, like 5, 7, 9 °C? Please clarify.

6) Line 118: . "For this, their original tubes were cut to 7 cm whereby care was taken to not harm the individual inside the tube". - Clarify why this was done (standardization ?) and if it can have any secondary effect? (Could this be a methodological limitation?)

7) Line 136-140 etc. I found a few spelling mistakes here and there, e.g. "new built" or "newly build"? (it should be "newly built") . The authors should check for grammar and typos carefully.

8) Building rate. Building rate (cm/h) was used to measure performance. I suppose length was measured per given time because of some variation in the duration of the experiments. However, this assumes that building rate does not vary through time, but that is not necessarily the case. An animal that has 2 hours to build its tube, may not have the same "average" building rate as an animal that was given 8 hours to build its tube. The authors should show the mean, SD and range of "building time" for all individuals and add "building time" as one of the covariates, to control for the duration of the building process. Even if there is no or little variation in building time (20 hrs, line 136), this should be clarified (mean and SD and range) and why certain units were used (length vs. length over time) should be explained and justified.

Reviewer's Responses to Questions

Experimental quality

Does each figure have the proper controls?

If 'No', please indicate reasons in Comments for Author box below.

Reviewer #1:

- Yes

Reviewer #2:

- Yes

Were the data analyzed using appropriate statistical tests?

If 'No', please indicate reasons in Comments for Author box below.

Reviewer #1:

- No

Reviewer #2:

- Yes

Reproducibility

Were experiments performed using adequate number of biological replicates?

If 'No', please indicate reasons in Comments for Author box below.

Reviewer #1:

- Yes

Reviewer #2:

- Yes

Does the methods section provide sufficient detail to permit reproducibility?

If 'No', please indicate reasons in Comments for Author box below.

Reviewer #1:

- Yes

Reviewer #2:

- Yes

Completeness

Are the manuscript's conclusions supported by the data?

If 'No', please indicate reasons in Comments for Author box below.

Reviewer #1:

- Yes

Reviewer #2:

- Yes

Scholarship

Do the authors cite and discuss the merits of data that would argue for and against their conclusion?

If 'No', please indicate reasons in Comments for Author box below.

Reviewer #1:

- Yes

Reviewer #2:

- Yes

Does the manuscript title & abstract accurately reflect the contents of the manuscript, without hyperbole?

If 'No', please indicate reasons in Comments for Author box below.

Reviewer #1:

- Yes

Reviewer #2:

- Yes

First revision

Author response to reviewers' comments

Reviewer 1: General remarks:

In this study, the authors generate a thermal performance curve for a species that does not lend itself readily to such investigations owing to its obligatory relationship with sandy sediments. I found the study to be an interesting contribution, especially since there was a nice correspondence
© 2026. Published by The Company of Biologists under the terms of the Creative Commons Attribution License (<https://creativecommons.org/licenses/by/4.0/>).

between the TPC and the temperature regime to which it is exposed in the field. While further studies on summer and winter acclimated animals would provide for a more complete characterisation of its TPC, the present study already shows that the methodology works for this species. My main suggestion is to do more with the data: Currently, data from two specimens is averaged and the analysis is focussed on tube length, but in addition to tube length, also tube weight and the mass of the animal itself are measured. Since this study is the first on this species it would perhaps be nice to analyse what performance measure is most appropriate (tube length, tube weight, % new tube mass, etc.) and explore whether the performance metric can be standardized by the mass of the animal.

Answer: Thank you for your positive and constructive comment. We indeed explored the performance metric but did not write about this exploration in the manuscript. We have changed that now. For this please see answers to your detailed remarks below.

Detailed remarks:

R1.1: Line 75: for a study explicitly studying the shape of the thermal performance curve across different levels of biological organisation see: <https://doi.org/10.1098/rstb.2018.0549>

A1.1: Thank you for pointing out a missing reference. We added your suggestion which fits well here (line 80) and added the reference to the reference list: Rezende, E. L., & Bozinovic, F. (2019). Thermal performance across levels of biological organization. *Philosophical Transactions of the Royal Society B*, 374(1778), 20180549. DOI: 10.1098/rstb.2018.0549

R1.2: Line 78: maybe say: "performance typically declines sharply". Note also that the paper cited (Portner's JEB review) (i) shows a more symmetrical unimodal curve, rather than a skewed performance curve which is more typical (e.g. see: <https://doi.org/10.1073/pnas.2513099122>), and (ii) different studies define the thermal maximum differently. Typically, T_{max} is the temperature at which performance becomes 0, and this is frequently equated to the upper thermal maximum, above which the animal dies from heat stress. See also: <https://doi.org/10.1073/pnas.0709472105>

A1.2: Done. We did not yet know about the interesting read from Arnoldi et al 2025. We exchanged the reference from Pörtner to Arnoldi et al. in line 84. The new reference nicely explains several competing mechanisms that try to explain the shape of the thermal performance curve so that we also bypass the debate about Pörtner's OCLTT hypothesis. Thank you.

The reference was also added to the reference list: Arnoldi, J. F., Jackson, A. L., Peralta-Maraver, I., & Payne, N. L. (2025). A universal thermal performance curve arises in biology and ecology. *Proceedings of the National Academy of Sciences*, 122(43), e2513099122. DOI: 10.1073/pnas.2513099122

R1.3: Line 93: I would say this study evaluates the thermal performance curve, rather than thermal tolerance (which is only the latter part of the thermal performance curve).

A1.3: Agreed. We have changed it accordingly (line 99).

R1.4: Line 144: Why was the average taken from the two individuals, rather than using both as separate data points (with possibly aquarium as a random factor to account for any effects)?

A1.4: While accounting for aquaria effect as a random factor is theoretically possible, non-linear models are already very complex and might be difficult to fit. For fitting TPCs we used the more widely used tool, package rTPC, that does not offer the option to include random effects. We decided to average responses instead of developing even more complex statistical methods because our main aim was to assess the experimental method and identify thermal limits of the species and because averaging responses within aquaria is not technically wrong. Measurements taken from the same aquarium are not independent and by averaging the two individuals we address this pseudo-replication (Hurlbert, 1984).

We added this information to the sentence and moved it to the Analysis section (line 264). We added the reference to the reference list: Hurlbert, S. H. (1984) Pseudoreplication and the Design of Ecological Field Experiments. *Ecological Monographs*, 54(2), 187-211. DOI: 10.2307/1942661

R1.5: Line 169: I would also include a figure of the temperatures to which these animals are likely exposed. It is highly interesting to see that the thermal range of the TPC is nicely bracketed by the lower and higher temperature ($\sim 7^{\circ}\text{C}$ and $\sim 21^{\circ}\text{C}$).

A1.5: Originally, we submitted this manuscript as a short communication for which we were restricted to the number of display items. We now moved the Supplementary Fig. S1 to the main

text of the article and included it as Fig. 2. It shows the measured mudflat temperatures in winter (January), spring (April), and summer (June) (line 404).

R1.6: Line 181: what was it about their physical state that prompted you to exclude said individuals? Also, I would place this information in the methods and focus in the results on those animals that were used on the analysis.

A1.6: Good suggestion to move this sentence to the methods section. We placed this information now in the methods (line 255). There were several reasons why we excluded these individuals, e.g. they did not show any signs of movement and we considered them dead, their soft tissue was injured, they were identified as a different species, their body weight could not be determined, or their tube was stuck between the aquarium walls too high above the sediment which prevented the individuals from being in contact with the sediment. We added these details in the methods section (line 255).

R1.7: Line 184: given that the mass of both the individual animals and that of the tube was measured, I was expecting an analysis where individual mass was included as a covariate. Also, the use of a control group is a great idea to test for variation in the condition of specimens collected on different days, so I was expecting an explicit test of this (and if there is no bias in the condition, then the control data could be added to the overall analysis).

A1.7: The species is an obligate tube dweller and removing it from its tube causes substantial stress to the individual. Therefore, we measured individual mass only at the end of the experiment, primarily to verify that we used individuals of similar size. Because experimental temperature may itself influence body mass, mass cannot be considered independent of treatment. Including it as a covariate would therefore risk controlling for a post-treatment variable and potentially removing part of the treatment effect. For this reason, mass was not included in the analysis. We added this information in the methods (line 265): “Newly built tube length was not standardized for individual size because body mass was determined only after the experiments. Because experimental temperature may influence body mass, mass could not be considered independent of treatment.”

Only one to two aquaria per day were assigned to the control temperature (Table S1), meaning that any test with “collection day” as a categorical variable would be unreliable due to low replication and therefore uninformative. We addressed any day-related bias by randomizing treatment temperatures across days, so that temperatures were evaluated in two or three different days, and so that every day a different set of temperatures was assessed in parallel while the control temperature was tested throughout the experimental days. This prevented collection day being a confounder of experimental temperature. We added a sentence in the methods section (line 238): “The control temperature (local condition, 11 °C) was used in one water bath each day to account for potential differences across experimental days (n = 8, Table S1). *All experimental temperatures were tested in two or three experimental days so that any differences in condition of individuals between days was randomized across temperatures (Table S1)*”.

R1.8: Line 195: Not sure what the function is of figure 2. Like I mentioned above, I would use an explicit analysis to test for differences across days in performance and then test for variation in tube building performance related to body mass. Then the response variable (mass of newly build tube) could be corrected for differences in mass and then this mass-corrected performance could be fitted with a thermal performance gaussian regression.

A1.8: The purpose of Fig. 2 (now Fig. 3 in the revised manuscript) is to illustrate the distribution of body mass data across days and temperatures, allowing readers to visually assess potential day effects and variation associated with body size. As outlined in our responses to comments R1.4 and R1.7, only one to two control aquaria were used per day, which makes formal tests for day effects unreliable due to low replication. To mitigate this, treatments were randomized across days (Table S1). Visual inspection of Fig. 3 indicates no evident patterns in body mass associated with day, particularly within the control group. Because body mass was only measured after the experiment and may itself be influenced by temperature, it cannot be treated as an independent covariate. We have added sentences in the Methods (as outlined in the responses to R1.4 and R1.7) to clarify these points and to highlight both the limitations of the current study and potential improvements for future experiments.

R1.9: Line 216: Not sure that I follow this rationale. The TPC is bracketed by the thermal minimal and thermal maxima currently experienced. So there is some local adaptation, but concluding that

these are spring acclimated and thus that the species will show a different TPC when summer acclimated is somewhat speculative with the data at hand. It could be that the species becomes less active during summer, but the temperatures in its habitat are not so extreme that they surpass the thermal range fitted here, right? I guess would could be better highlighted as spring acclimation is the peak in the TPC which coincides exactly with the temperature experienced.

A1.9: We kindly disagree with this comment. Temperatures in the field later during the year e.g. summer, are well above the upper thermal tolerance fitted here. We visualize this in Figure 2 which we now moved from the supplementary material to the main text. It shows the measured mudflat temperatures in winter (January), spring (April), and summer (June). In paragraph 2 (line 142) and 3 (line 159) of the discussion, we further elaborate on our rationale why we suggest that the thermal coping range measured in our study reflects spring acclimation rather than fixed thresholds.

R1.10: Line 301: The doi link is not working.

A1.10: Apologies. The DOI will only be accessible upon publication of this manuscript. We now created a link that gives you access to the draft version of the zenodo entry. Link: https://zenodo.org/records/17715259?preview=1&token=eyJhbGciOiJIUzUxMiJ9.eyJpZCI6IjNiNTA1Nzk4LTA4MDktNDQ2ZC04NmNlLThiYTA5MDQ0YjA2YyIsImRhdGEiOiJ9LCJyYW5kb20iOiI2MjZmMWE5NjAyYTE3NTBjY2E4NWNjYTg5Y2JjYWM5ZCJ9.QC0hTJdw19LVBbin_aUHO94rOkxnh1OdbdLVGSnMtZAzz8K_WejavVwAPopbtn5ioui3Gf6Wa5sjgIJ9erVA

Reviewer 2:

General comments

This paper aims at testing the effect of temperature on a habitat-forming species, the tube worm *Lanice conchilega*. Because standard methods are unsuitable to measure this species' performance, the authors devised a simple method that takes into account the ability of this species to form their tube. The authors found that tube-building activity peaks at around 12 °C, with minimum and maximum at around 3 and 21 °C, respectively. Tube-building by this species is an activity of great ecological importance, since it modifies sediments and habitat structure, and can positively affect species richness. Therefore, by devising a simple methodology to measure tube-building performance and by assessing the temperature performance curve of this key species, this paper provides a potentially useful contribution to the literature on the effect of global warming on intertidal communities. I only have a few minor comments.

Answer: Thank you for your kind words!

Specific comments

R2.1: 1) The figures are clear and the experiments were appropriately designed, including a control. The statistics are appropriate.

A2.1: Thank you

R2.2: 2) The authors conclusions are supported by the data. The paper's limitations section is also well written and the use of only one season is listed as one of the limitations. (Line 93). The authors wrote: "Because our experiment was conducted with spring-acclimatized individuals, it does not allow direct inference about the species' winter sensitivity (de la Barra et al., 2025)." This is true for the winter but also of any other season.

A2.2: We fully agree with the reviewer. We refer to the winter sensitivity of the species because it is often mentioned in the published literature. We continue the line of thought by writing (line 102) "It provides a baseline understanding of its temperature-dependent tube-building ability under spring conditions that can be extended to other seasons" which we believe addresses the reviewers' comment.

R2.3: 3) The state of the art is cited appropriately.

A2.3: Thank you

R2.4: 4) A new, simple method to assess performance is carefully described.

A2.4: Thank you

R2.5: 5) line 116: "To determine the thermal performance of *L. conchilega*, individuals were exposed to every other temperature between 5 and 23 °C for 20 h." Not clear what temperatures the authors refer to. You mean every other degree, like 5, 7, 9 °C? Please clarify.

A2.5: That is exactly what we meant. To be more precise we rewrote the sentence to (line 225): "The thermal performance of *L. conchilega* was determined by exposing different individuals for 20 h to one of ten test temperatures ranging from 5 to 23 °C at 2 °C intervals. After the temperature exposure, the length of the newly built tube was measured (Fig. 1)."

R2.6: 6) Line 118: "For this, their original tubes were cut to 7 cm whereby care was taken to not harm the individual inside the tube". - Clarify why this was done (standardization?) and if it can have any secondary effect? (Could this be a methodological limitation?)

A2.6: The tube lengths varied in size when collected from the field and one side had a fringe of variable size that collapsed during collection. To standardize tube size and structure at the start of the experiment, we cut all tubes to 7 cm, which was large enough for the worms to hide in. This ensured consistent starting conditions, but we cannot fully exclude minor effects on the worm. Yet, all individuals were treated identically, so any such effects would not bias comparisons between treatments. We modified the original sentence and moved it to the "Species collection section". It now reads (line 218): "The tube lengths varied in size and one side had a fringe of variable size that collapsed during collection. To standardize tube size and structure at the start of the experiment, we cut all tubes to 7 cm whereby care was taken to not harm the individual inside the tube. The tube length was large enough for the worms to hide in and this procedure ensured consistent starting conditions".

R2.7: 7) Line 136-140 etc. I found a few spelling mistakes here and there, e.g. "new built" or "newly build"? (it should be "newly built"). The authors should check for grammar and typos carefully.

A2.7: Thank you for pointing this out. We corrected the word in line 250 and 401.

R2.8: 8) Building rate. Building rate (cm/h) was used to measure performance. I suppose length was measured per given time because of some variation in the duration of the experiments. However, this assumes that building rate does not vary through time, but that is not necessarily the case. An animal that has 2 hours to build its tube, may not have the same "average" building rate as an animal that was given 8 hours to build its tube. The authors should show the mean, SD and range of "building time" for all individuals and add "building time" as one of the covariates, to control for the duration of the building process. Even if there is no or little variation in building time (20 hrs, line 136), this should be clarified (mean and SD and range) and why certain units were used (length vs. length over time) should be explained and justified.

A2.8: Thank you for this thoughtful comment, which highlights an aspect that we had not yet considered. We initially showed tube-building performance as length per hour (cm h⁻¹), as standardizing responses per hour is common practice and the unit cm per 20 h appeared unintuitive. However, we agree that tube-building rate may vary over time similar to its irregular oxygen-pumping behaviour (line 90). We have therefore revised the analysis to report newly built tube length over the entire building period (20 h). We updated the results section (line 125) and Fig. 4 accordingly (line 420). In addition, we have added the following sentence to the Methods section (line 268): "Newly built tube length was analyzed as the cumulative length produced over the experimental period, as experimental duration was identical across treatments and experimental runs (20 ± 0 h; mean ± s.d.) and tube-building activity was not assumed to be constant over time."

Second decision letter

MS ID#: bio.062398R1

MS Title: Thermal window of exercise performance of the ecosystem engineer *Lanice conchilega*

Authors: Katharina Alter; Nienke Zwaferink; Paula de la Barra

Dear Dr Alter,

I've had the time this morning to read through your rebuttal and the associated changes you've made to your manuscript, and I am happy to tell you that your manuscript has now been accepted for publication in Biology Open, pending our standard publication integrity checks. It was accepted on 19 Dec 2025.